# Development and Application of a 10 kV Mechanical DC Circuit Breaker

**Lu Qu** [1,2,*][ID]**, Zhanqing Yu** [1,2]**, Xiang Xiao** [3]**, Wei Zhao** [3]**, Yulong Huang** [1,2] **and Rong Zeng** [1,2,*]

[1]  State Key Lab of Power Systems, Department of Electrical Engineering, Tsinghua University, Beijing 100084, China; yzq@tsinghua.edu.cn (Z.Y.); yulonghuang@mail.tsinghua.edu.cn (Y.H.)
[2]  DC Research Center, Energy Internet Institute, Tsinghua University, Beijing 100084, China
[3]  Electric Power Research Institute, Guangdong Power Grid Co., Ltd., Guangzhou 510080, China; xiaoxiang@dky.gd.csg.cn (X.X.); zh_solar@126.com (W.Z.)
*  Correspondence: qulu@tsinghua.edu.cn (L.Q.); Zengrong@tsinghua.edu.cn (R.Z.); Tel.: +86-183-5602-7135 (L.Q.)

**Abstract:** A DC circuit breaker is piece of core equipment for DC grid construction and can achieve fast isolation of DC faults in the grid. In this paper, based on the fault characteristics and protection requirements of an AC/DC hybrid distribution network, the technical parameters and topology structure of an inductance and capacitance (LC) resonant commutation-type mechanical DC breaker are proposed. Furthermore, an engineering prototype of the mechanical DC circuit breaker is developed, then a small current breaking test and a large current breaking test are carried out. The test results show that a 10 kA fault current can be interrupted within 10 ms by the engineering prototype. Finally, the engineering prototype of the mechanical DC circuit breaker is installed in the AC/DC hybrid distribution network project, and a current breaking test is carried out. The test results show that the mechanical DC circuit breaker has good running conditions, and the function and performance meet the engineering design and operation requirements.

**Keywords:** mechanical DC circuit breaker; LC resonant commutation; DC distribution network

## 1. Introduction

A flexible DC grid has the advantages of no need for reactive power compensation equipment, no commutation failure, and a power supply for passive systems, thus it has become one of the hot research directions in recent years [1–3]. An important challenge for flexible DC grids is the ability to clear DC-side faults quickly. A DC circuit breaker can realize fast fault isolation and clearance, thereby avoiding the overall shutdown of the multi-terminal DC system, which can effectively achieve flexible networking and improve the stability of operation. Therefore, the engineering applications of the DC circuit breaker are of great significance to the development of flexible DC grids [4,5].

At present, DC circuit breakers can be divided into: mechanical DC circuit breakers, solid state DC circuit breakers, and hybrid DC circuit breakers [4]. Among them, mechanical circuit breakers have low on-state loss and low cost, which has become a development direction for DC circuit breakers [6–9]. There are two main methods for mechanical DC circuit breakers to achieve DC fault current breaking by manufacturing manual zero-crossing points [10,11]. The first method is to use the arc voltage generated by the mechanical switch during the breaking process to achieve current zero crossing. In order to reduce the arcing time and reduce the influence on the ablation of the contacts, it is required that the arc voltage must be established quickly in the initial stage of the breaking and must be higher than the system voltage, so this method is only applicable to lower voltage levels [12,13]. The second method is to use the resonant current generated by the LC resonant commutating branch, superimposing it

on the fast mechanical switch branch, causing its current to quickly cross zero and extinguish the arc, while transferring the current of the fast mechanical switching branch to the LC resonant commutation branch. Compared to traditional mechanical circuit breakers, this LC resonant commutation type mechanical DC circuit breaker has higher breaking speed and shorter arcing time, and has been widely researched and applied in recent years [14–17].

In this paper, the technical parameters of the mechanical DC circuit breaker are given for the fault protection of a 10 kV AC/DC hybrid distribution network. Then, a topology based on the LC resonant commutation type mechanical DC circuit breaker is proposed, the engineering prototype is developed, and a DC current breaking test is carried out. Finally, the engineering prototype of the mechanical DC circuit breaker is installed in the AC/DC hybrid distribution network project to achieve fast isolation of DC faults, thus ensuring safe and reliable operation of the system.

## 2. Project Overview

In order to effectively support the access of high permeability of renewable energy and the supply of high proportion of DC load, the AC/DC hybrid distribution network is constructed by two four-port DC transformers. The basic topology is shown in Figure 1.

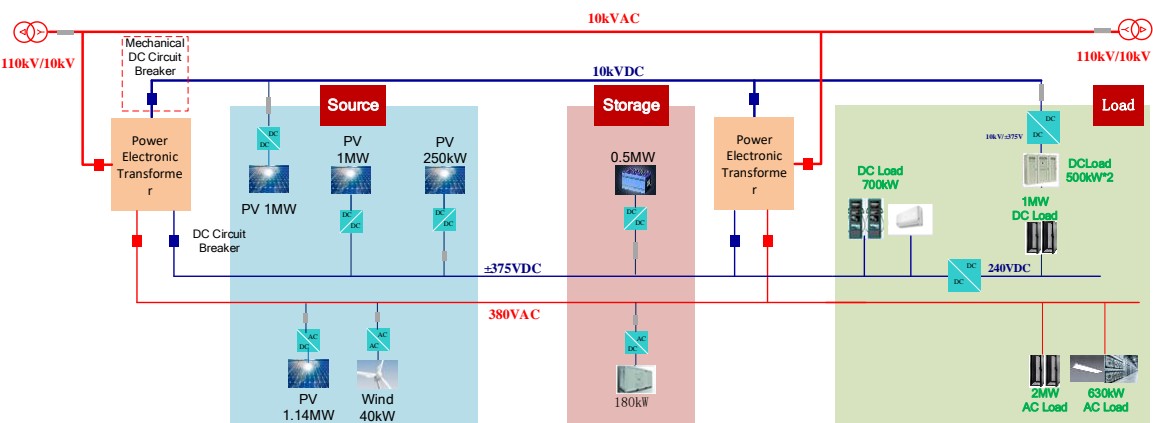

**Figure 1.** Topology of AC/DC hybrid distribution network.

The system includes 10 kV AC and 10 kV DC distribution systems, and 380 V AC and ±375 V DC power supply systems. The type and capacity of source, storage, and load in the 10 kV AC/DC hybrid distribution network are shown in Table 1. In order to improve the efficiency of the renewable energy access system, 1 MW photovoltaic is connected to a 10 kV DC bus in centralized mode, and 1 MW photovoltaic is connected to a ±375 V DC bus in distributed mode.

**Table 1.** Type and capacity of source, storage and load in a 10 kV AC/DC hybrid distribution network.

| Voltage | Source | Storage | Load |
|---|---|---|---|
| 10 kV DC | PV 1 MW | — | 500 kW × 2 Industrial DC load |
| ±375 V DC | PV 1 MW<br>PV 250 kW | 500 kW | 150 kW Office DC load<br>550 kW DC load<br>1 MW DC IT load |
| 380 V AC | PV 1.14 MW<br>Wind 40 kW | 180 kW | 2 MW AC IT load<br>630 kW AC load |

In this system, the DC load is 2.7 MW, accounting for 50.66% of the total load. Among them, 1 MW DC load is a resistor for a temperature rise test, which is connected to the 10 kV DC bus by a DC/DC converter. 150 W DC load is the DC air conditioner, which is connected to the ±375 V DC bus by a DC/DC converter. 550 W DC load is the DC charging pile, which is connected to the ±375 V DC

bus by a DC/DC converter. 1 MW DC load is the information technology (IT) load, which is connected to the 240 V DC bus by a DC/DC converter.

The structure has the following advantages: (1) the reliability and economy of the operation can be improved by the networking mode of the power electronic transformer; (2) according to the capacity of the renewable energy source, a power grid with different voltage levels can be flexibly selected for cost saving and loss reduction and; (3) a two-station combined loop operation can be realized by 10 kV DC to improve power supply reliability, and at the same time, as a power traversing channel, load balancing is realized.

## 3. Parameter Configuration of Circuit Breaker

In order to improve the fault clearing capability of the system, for this section we will carry out a transient simulation and analysis of system faults, and provide reference for the selection of technical parameters of mechanical DC breakers.

There are three main types of DC-side faults in AC/DC hybrid distribution networks, namely, line breakage faults, pole-to-ground short-circuit faults, and inter-pole short-circuit faults [18]. Among them, an inter-pole short-circuit fault can cause the most serious damage to the system. Therefore, an inter-electrode short-circuit fault is taken as an example for simulation analysis.

When a 10 kV inter-pole short-circuit fault occurs, the fault characteristics of the DC port are mainly affected by the side close to the fault point in the dual active bridge (DAB) module, due to the electrical isolation of the high frequency transformer in the DAB module of a DC transformer. The one-side structure of the DAB module in a DC transformer is shown in Figure 2, where CDAB is the outlet capacitor of the DAB module and VDC is the voltage of the capacitor terminals. L is the outlet inductance of the DAB module, and IDC represents the short circuit current flowing through the inductance. VD is the reverse parallel diode of Insulated Gate Bipolar Transistor (IGBT) in the H bridge, and $R_k$ is the transition resistance of the inter-pole short circuit.

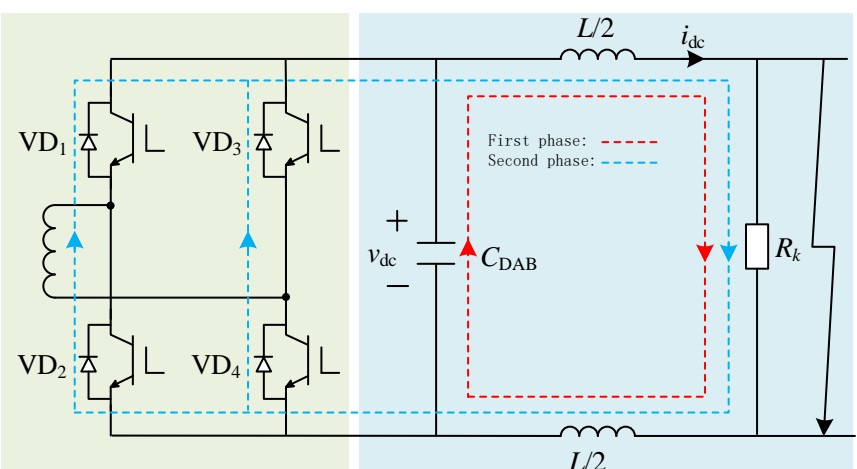

**Figure 2.** Two stages of a short circuit current in the event of short circuit between poles.

The first zero-crossing point of capacitance voltage is the demarcation point between the resistance inductance and capacitance (RLC) resonance stage and inductance continuation stage, at which the maximum short-circuit current is flowing through inductance is [19],

$$I_{\text{dcmax}} = \frac{V_{\text{dc}}}{\omega_0 L} \tag{1}$$

where $\omega_0 = 1/\sqrt{LC_{\text{DAB}}}$. Assume that the peak value of the fault current is 2 kA, that is, $I_{\text{dcmax}}$ = 2 kA, and $V_{\text{DC}}$ = 10 kV, $C_{\text{DAB}}$ = 100 uF is substituted into Equation (1), and $L$ = 2.5 mH can be obtained. In order to analyze the effect of inductance on the fault current limit, the outlet inductance of the DAB

module is set to 2 mH, 6 mH, 10 mH, 14 mH, and 18 mH, and the fault current characteristics are shown in Figure 3.

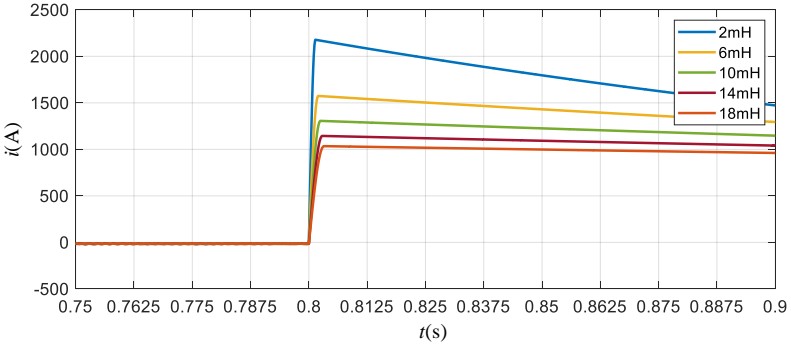

**Figure 3.** Fault current at a 10 kV inter-pole short-circuit fault.

The peak value of the fault current is shown in Table 2.

**Table 2.** Peak value of the fault current at a 10 kV inter-electrode short-circuit fault.

| Reactance Value (mH) | 2 | 6 | 10 | 14 | 18 |
|---|---|---|---|---|---|
| Peak value of fault current (A) | 2178 | 1574 | 1306 | 1144 | 1034 |
| Peak time (ms) | 1.404 | 2.005 | 2.465 | 2.880 | 3.256 |

According to the results of the fault simulation, considering the factors of the breaking current, breaking time, cost, and other factors, the technical parameters of the mechanical DC circuit breaker can be given, as shown in Table 3.

**Table 3.** Technical parameters of the DC circuit breaker.

| Parameter | Unit | Value of Parameter |
|---|---|---|
| Rated DC voltage | kV | 10.0 |
| Rated DC current | A | 1000 |
| Maximum continuous DC current | A | 1100 |
| Break time | ms | 10 |
| Rated breaking current | kA | 10 |
| Rated DC withstand voltage, to ground | kV | 16 |
| Rated operational impulse withstand voltage peak, to ground | kV | 20 |
| Rated lightning impulse withstand voltage peak, on the ground | kV | 24 |

## 4. Plan and Development

### 4.1. Topological Structure

A typical topological structure of a mechanical DC circuit breaker based on LC resonant commutation is shown in Figure 4, wherein M refers to the quick mechanical switch, S1–S5 refers to the thyristor, K refers to the bypass switch, *C* refers to the resonant capacitor, *L* refers to resonant inductance, $R_p$ refers to static equalizing resistance, $R_s$ refers to dynamic equalizing resistance, $C_s$ refers to dynamic equalizing resistance, MOA1–MOA2 refers to the metal oxide arrester for restricting over-voltage, and MOA refers to the metal oxide arrester for restricting over-voltage and fault energy absorption.

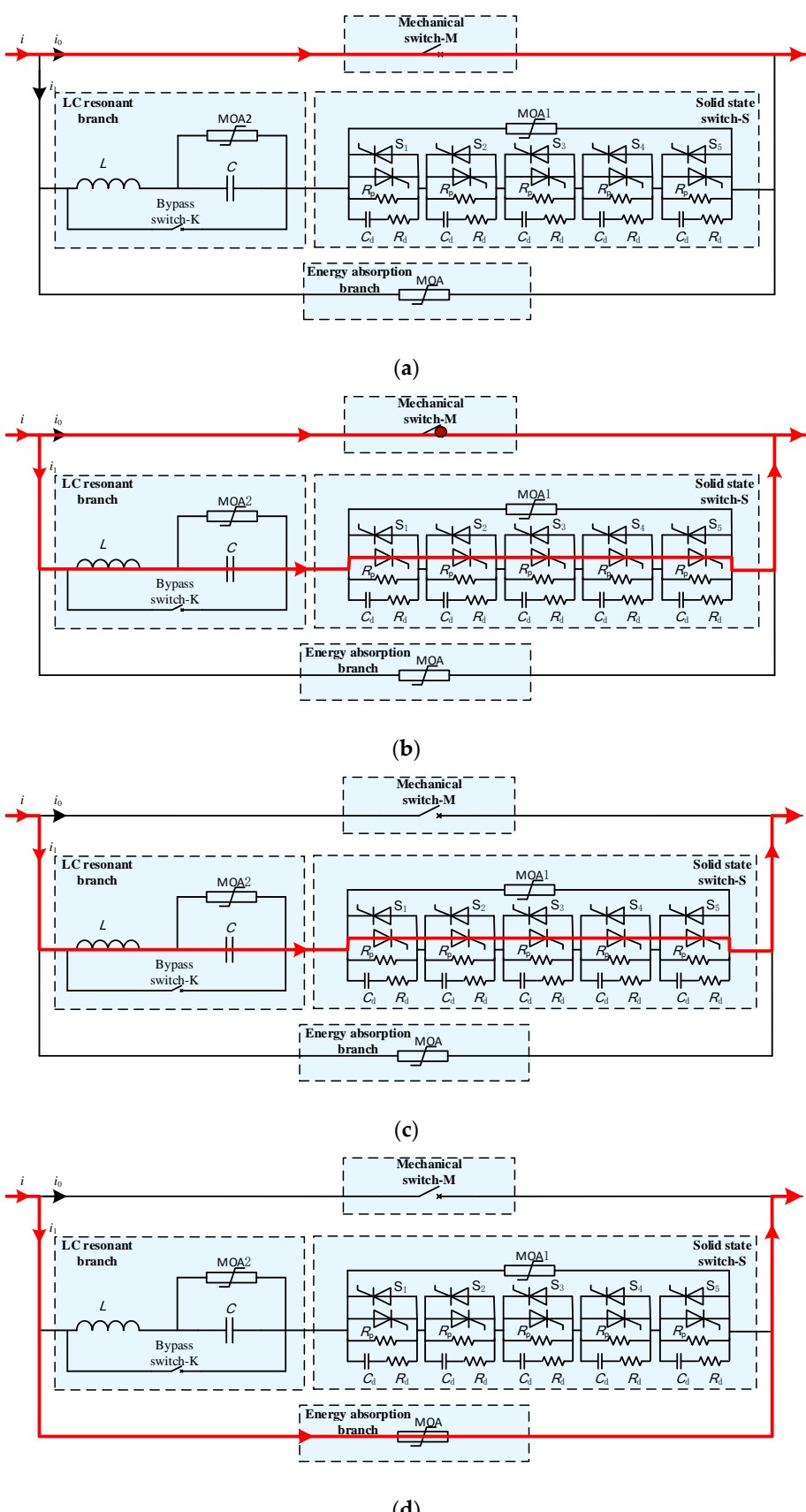

**Figure 4.** Topological structure and breaking process of a mechanical DC circuit breaker. (**a**) The flow-through process of the fast mechanical switch branch; (**b**) the converting process from the fast mechanical switch branch to the LC resonant commutation branch; (**c**) The flow-through process of LC resonant commutation branch; (**d**) the energy dissipation process of system failure.

In normal operation, the fast mechanical switch M is closed, and the solid state switch S and the bypass switch K are disconnected. The fast mechanical switch branch undertakes the normal current flow, and the resonant capacitor C in the LC resonant commutation branch is pre-charged. When a fault occurs, the control and protection system of the mechanical DC circuit breaker sends the switching command to the fast mechanical switch. After a certain mechanical delay, the fast mechanical switch starts to open and generates arcs between the dynamic and static contacts. When the contact distance reaches the safety gap, the solid state switch of the LC resonant commutation branch is turned on. The resonant current generated by the LC resonant branch is superimposed on the fast mechanical switch branch, and the fast mechanical switch crosses zero to extinguish the arc. At the same time, the fault current is rapidly transferred from the fast mechanical switch branch to the LC resonant commutation branch. Subsequently, the fault current charges the resonant capacitor C, and the voltage at both ends of the fast mechanical switch increases continuously. When the gap of the fast mechanical switch moves, it can withstand the transient recovery voltage, and the solid state switch S is turned off. At this time, the transient recovery voltage reaches the on-line voltage of the MOA, and the fault current is rapidly transferred from the LC resonant commutation branch to the energy dissipation branch. Then, the MOA absorbs the fault energy of the system, and the fault current rapidly drops to zero. The whole interruption process is completed.

## 4.2. Prototype Development

The development of the mechanical DC circuit breaker mainly includes a fast mechanical switch, solid state switch, LC resonant circuit, and control and protection system.

### 4.2.1. Development of the Fast Mechanical Switch

Because of the fast speed of the solid state switch, the speed of the fast mechanical switch will directly affect the breaking speed of the mechanical DC circuit breaker. Therefore, the arc extinguishing mode and operating mechanism of the fast mechanical switch are selected and designed to ensure that the fast mechanical switch can quickly and reliably interrupt the fault current.

Compared to other operating mechanisms, the permanent magnet operating mechanism has a simple structure, few components, and low cost. At present, the opening time of a vacuum switch using a permanent magnet operating mechanism is generally several milliseconds to several tens of milliseconds. Therefore, the operating mechanism of the fast mechanical switch selects the permanent magnet operating mechanism. Compared to asulfur hexafluoride medium, the insulation recovery speed of the vacuum medium after arcing is faster, and it has the characteristics of light contact quality, short overtravel, and a small opening distance, which makes it easy to realize rapid opening and closing. Therefore, the arc extinguishing mode of the fast mechanical switch selects the vacuum interrupter.

Based on the permanent magnet operating mechanism and the vacuum interrupter, a 10 kV fast electromagnetic mechanical switch is developed and tested for breaking. The results are shown in Figure 5.

It can be seen from the test results that the fast mechanical switch uses a larger discharge capacitor to improve the opening speed, and the higher the charging voltage of the discharge capacitor, the faster the opening speed. The final determination of the discharge capacitor is 106μF, the charging voltage is 800 V, and the opening time is 6.7 ms.

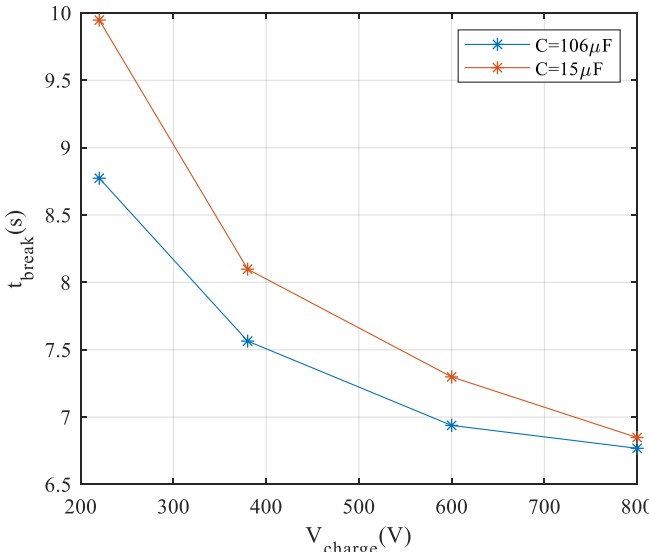

**Figure 5.** Test results of fast mechanical switch.

### 4.2.2. Development of the Solid State Switch

Traditional mechanical DC circuit breakers use high voltage spherical gap switches to control the discharge of the resonant capacitor C in the commutation branch, while high voltage spherical gap switches will generate sparks during the turn-on period, which can threaten the oil insulation equipment around the switch, and also lead to trigger failure, zero-crossing arc extinguishing, and ablation problems, thus weakening the reliability of interruption.

In order to avoid above problems, this paper uses a solid state switch based on power electronics to control the discharge of the resonant capacitor. Compared to a spherical gap switch, a solid-state switch has a fast breaking speed (tens μs~hundreds μs), due to the use of power electronic devices. Since it has no switching gap, its breaking reliability is high, and combined with the control and protection system, it is more controllable.

### 4.2.3. Development of the LC Resonant Circuit

For the LC resonant branch of the mechanical DC circuit breaker, the higher the resonant frequency $f$ is, the greater the $di/dt$ before the current of the fast mechanical switch crosses zero, and the higher the breaking capacity of the fast mechanical switch required. However, the resonant capacitance and resonant inductance of the corresponding LC resonant branch will decrease, which is beneficial to reducing the overall cost of the DC circuit breaker. The lower the resonant frequency $f$, the higher the breaking time of the fast mechanical switch, the higher the resonant capacitance and inductance, and the higher the switching cost. In addition, when the fault current is reversely interrupted, the first half wave of the LC resonant current will be superimposed on the current of the fast mechanical switch, and the lower the resonant frequency is, the more energy is injected into the arc, but this is not conducive to the recovery of the dielectric strength after the arc of the fast mechanical switch. Previous studies have shown that a resonant frequency $f$ of several kHz is appropriate [20]. Considered comprehensively, the resonant frequency of the LC resonant branch of the device is 2.5 kHz, which can improve the breaking speed on the one hand, and reduce the demand for dynamic breaking characteristics of solid-state switches on the other hand. When the resonant frequency $f$ is selected, the charging voltage of the capacitor is usually reduced by increasing the resonant capacitance $C$ and decreasing the resonant inductance $L$, while the frequency and amplitude of resonant current are guaranteed. Considering the cost, breaking time, breaking current and operation reliability of circuit breaker, $L = 20$ μH, $C = 200$ μF and $V_c = 5$ kV are selected.

### 4.2.4. Development of the Energy Dissipation Branch

The energy dissipation branch is mainly used for the absorption of fault energy and the overvoltage protection of the solid state switch. A metal oxide arrester (MOA) is used, and the design parameters are shown in Table 4.

**Table 4.** Design parameters of metal oxide arrester.

| Parameter | Value |
|---|---|
| Rated voltage | 10 kV |
| 1 mA reference voltage | 11.5 kV |
| 15 kA operating residual pressure | 16 kV |
| Absorbed energy | >500 kJ |
| Climbing distance | 25 mm/kV |

Among them, the 15 kA operating residual pressure should be less than the overvoltage level of the solid state switch. The absorbed energy depends on the fault energy of the AC/DC hybrid power distribution system.

### 4.2.5. Development of the Control and Protection System

The control and protection system is one of the most important components of a mechanical DC circuit breaker. It is mainly divided into four parts: control unit, control sub-unit, body overcurrent protection unit, and intelligent interlocking device. The basic configuration is shown in Table 5. The control and protection system receives motion commands from the upper system and sends them to the fast mechanical switch and solid switch to meet the functional requirements of the DC circuit breaker. It also has the body protection function of a DC circuit breaker to ensure the safety of the DC circuit breaker under extreme fault conditions.

**Table 5.** Configuration scheme of a control and protection system for a DC circuit breaker.

| Device Name | Function |
|---|---|
| Control unit | Complete data interaction with upper layer security devices and monitoring |
| Control sub-unit | Receiving control commands from the control unit to control the DC breaker |
| Body overcurrent protection unit | Complete current sampling / body overcurrent protection of the circuit breaker |
| Intelligent interlocking device | Realize interlocking of mechanical switch and grounding switch |

In order to verify the function and performance of the mechanical DC circuit breaker, a sequential control test, operation test, and breaking test should be carried out, and all control strategies performed by the control and protection system.

### 4.2.6. Development of the Structure

Based on the design results of the above-mentioned fast mechanical switch, the solid state switch and the LC resonance branch, the structural design of the DC circuit breaker is carried out. The mechanical DC circuit breaker prototype is divided into three cabinets, which are the grounding switch cabinet, the circuit breaker body, and the control cabinet, as shown in Figure 6.

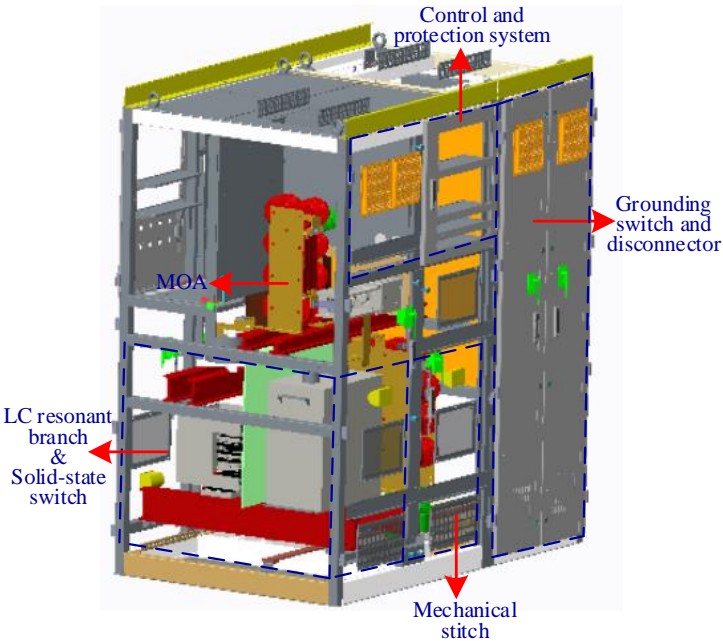

**Figure 6.** Structure of the mechanical DC circuit breaker.

### 4.3. Breaking Test

The breaking test of the mechanical DC circuit breaker adopts the LC oscillation mode, and the test circuit is shown in Figure 7. After charging the bus capacitor (Cs) with the DC source (U), the charging switch (K0) is turned off, the closing switch (K2) is closed, and the measured switch is turned off when the current reaches the first peak.

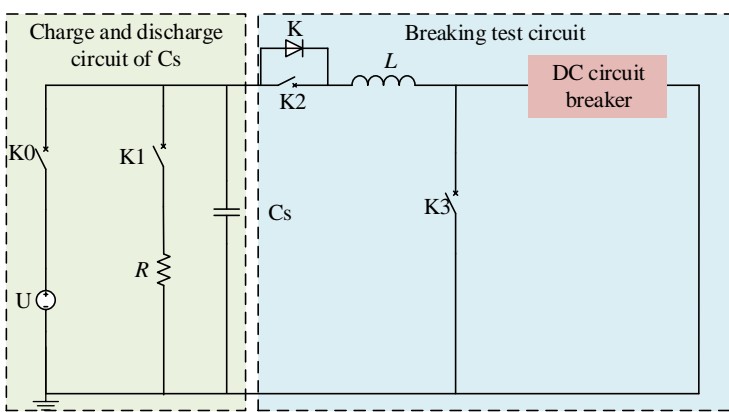

**Figure 7.** Circuit configuration for breaking test.

### 4.3.1. Small Current Breaking

The maximum breakable fault current based on the LC resonant commutated mechanical DC circuit breaker is the peak value of the LC resonant current, as shown in Figure 8. During the LC resonance process, the LC resonant current injected into the mechanical switch branch reaches the current peak after a quarter of the resonant period, and the current peak will be reversely superimposed with the fault current on the fast mechanical switch branch, so that the mechanical switch branch current crosses zero. If the starting currents of the two are opposite, then the zero crossing is superimposed in the opposite direction for three quarters of the period. At this moment, the capacitance voltage of the resonant capacitor C is zero. According to the topology diagram of the circuit, the transient recovery voltage generated by the zero-crossing of the fast mechanical switch is determined only by $L\mathrm{d}i/\mathrm{d}t$.

Since the resonant current reaches a peak value, $L di/dt$ is zero, so the transient recovery voltage (TRV) experienced by the mechanical switch during vacuum breaking is also zero.

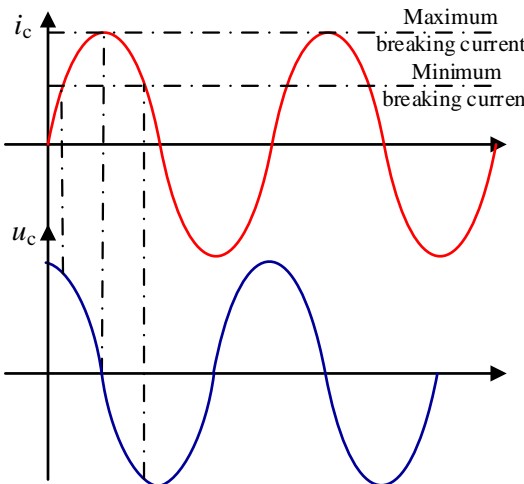

**Figure 8.** Current and voltage curves of resonant capacitors.

On the surface, the mechanical switch branch is more likely to cross zero at low currents, making it easier to break. However, unlike breaking a large current, the breaking of a small current will cause the mechanical switch to withstand a large transient recovery voltage. As shown in Figure 8, there are two intersections between the small current and the first half of the resonant current, that is, there are two "zero crossings" in the main branch, and the voltage of the resonant capacitor and the voltage of the resonant inductor corresponding to the "zero crossing" moment are not zero, at this time, the transient recovery voltage generated by the mechanical switch breaking can be described by Formula (2):

$$U_{VI} = L\frac{d_i}{d_t} + u_C \qquad (2)$$

Since the fault current is small, its transient recovery voltage becomes twice the capacitance voltage, and its overvoltage level threatens the breaking reliability of the mechanical switch. If the transient recovery voltage exceeds the rated withstand voltage of the mechanical switch, the mechanical switch will not be able to effectively break and the small current will fail to clear, so the small current breaking condition brings the risk of breaking failure to the DC circuit breaker.

Therefore, a 40 A small current forward and reverse breaking test is performed for the developed mechanical DC circuit breaker, and the test waveform is as shown in Figure 9. In the figure, trigger is the action trigger signal of the DC breaker, $I_{all}$ is the line current, $I_{cb}$ is the current of the mechanical switch branch, and $I_{Lc}$ is the current of the resonant branch.

It can be seen from the test results that when the current rises to 40 A, the mechanical switch operates, and after 9.0 ms, the fault current crosses zero, and the mechanical switch is turned off. Then, the current is transferred to the solid-state switching branch. After 0.8 ms, the solid-state switch is turned off, the fault energy is transferred to the MOA branch, and the breaking process ends.

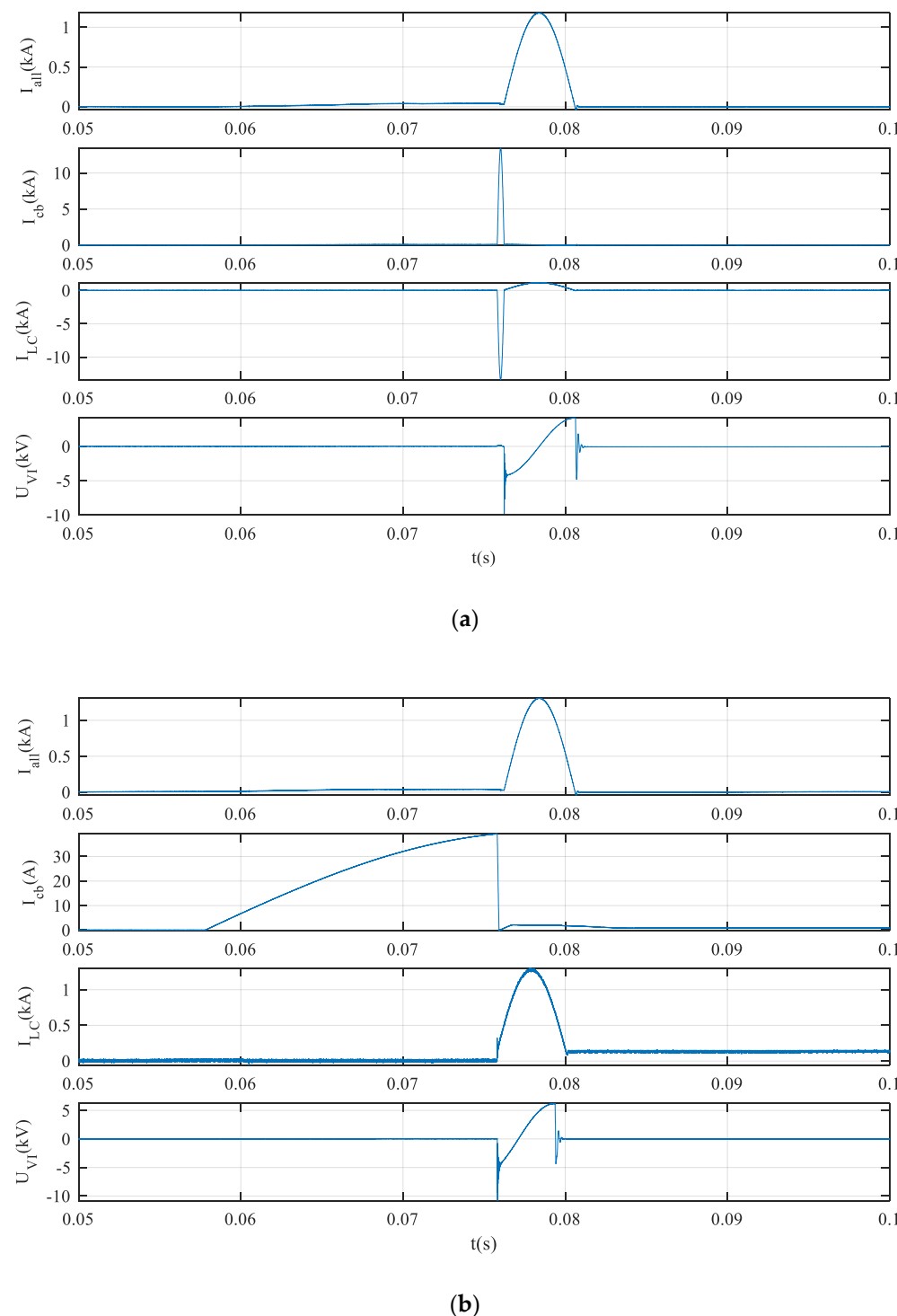

**Figure 9.** Small current breaking waveform. (**a**) 40 A small current forward breaking; (**b**) 40 A small current reverse breaking.

### 4.3.2. Large Current Breaking

In order to verify that the mechanical DC circuit breaker meets the design parameters, a 10 kA large current forward and reverse breaking test are performed. The test waveforms are shown in Figure 10.

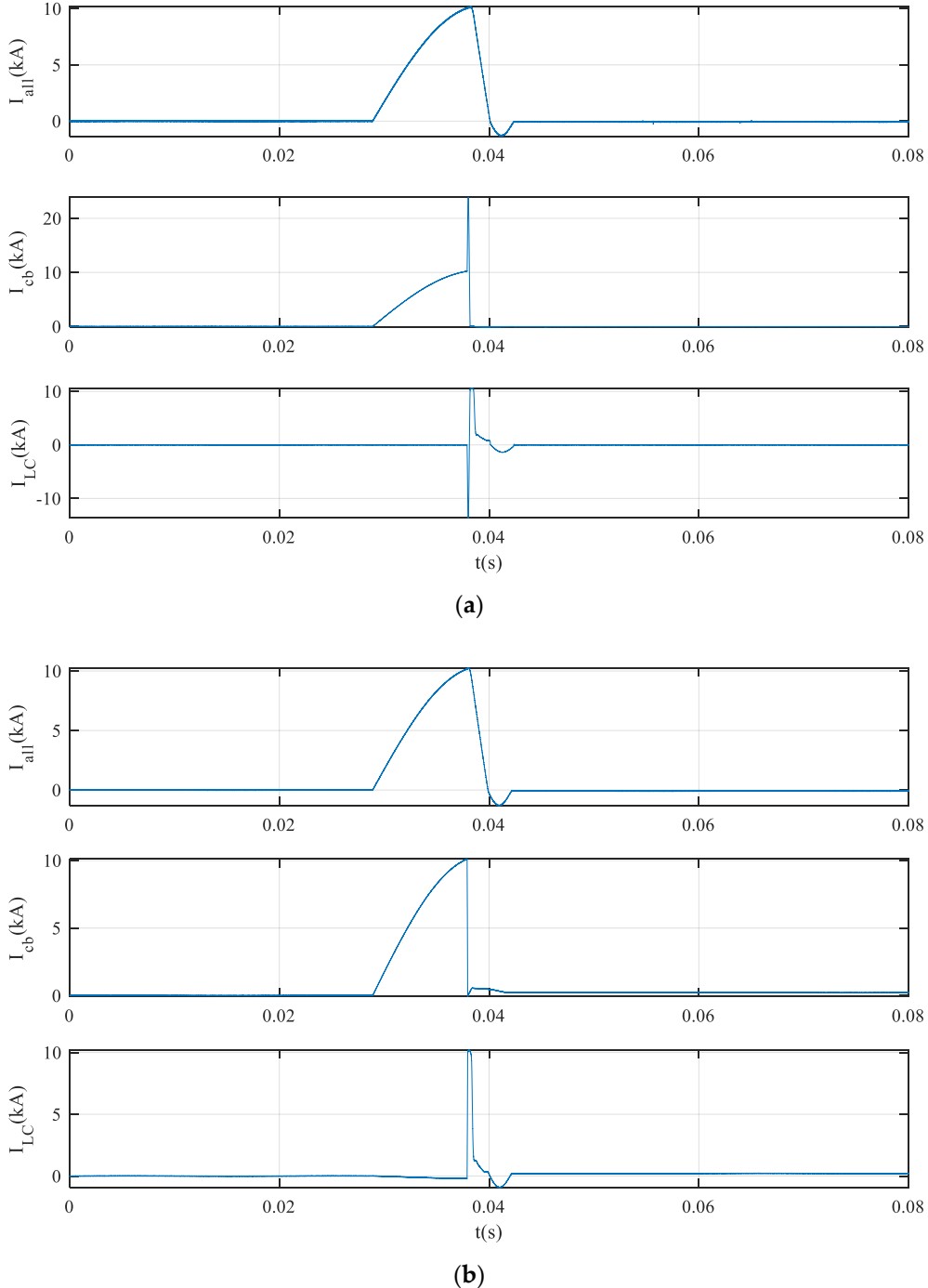

**Figure 10.** Large current breaking waveform. (**a**) 10 kA small current forward breaking; (**b**) 10 kA small current reverse breaking.

It can be seen from the test results that when the current rises to 10 kA, the mechanical switch operates, and after 8.3 ms, the fault current crosses zero, and the mechanical switch is turned off. Then, the current is transferred to the solid-state switching branch. After 0.8 ms, the solid-state switch is turned off, the fault energy is transferred to the MOA branch, and the breaking process ends.

Through the development and testing of the prototype of the mechanical DC circuit breaker, the feasibility of the overall design scheme is verified. The design parameters can meet the requirements of fault protection for the AC/DC hybrid power distribution system, thus laying the foundation for the engineering applications of the mechanical DC breaker.

## 5. Project Application

*5.1. Installation Plan*

In order to verify the function and performance of the mechanical DC circuit breaker, it is installed in the AC/DC hybrid distribution network, thereby achieving reliable isolation of the fault, and ensuring the safe and stable operation of the system. The schematic diagram of the installation is shown in Figure 11.

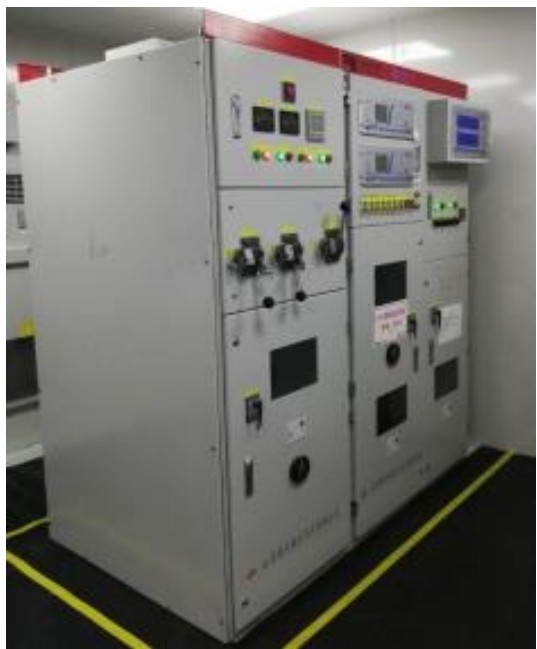

**Figure 11.** Installation diagram of the 10 kV mechanical DC circuit breaker.

*5.2. Breaking Commissioning*

In order to verify the function and performance of the mechanical DC circuit breaker, a sequential control test, operation test, and breaking test of the circuit breaker are carried out in the AC/DC hybrid distribution network project. The following mainly introduces the current breaking test.

For the operational requirements of the AC/DC hybrid distribution network demonstration project, the control strategy of the current breaking for the AC/DC hybrid power distribution system is as follows:

(1)   Imitating a 10 kV DC bus failure;
(2)   The control and protection system judges the faults and sends an interruption command tge to DC circuit breakers;
(3)   DC circuit breaker breaks current to realize fault isolation.

The test waveform of the system test is shown in Figure 12.

The test results show that the short-circuit fault occurs at the exit of the power electronic transformer at 64 ms. After the 3 ms protection exit time, the mechanical DC breaker performs the breaking action, and the 25 A current is successfully interrupted after 8.3 ms. This test verifies that the mechanical DC circuit breaker can realize the fault current breaking of the AC/DC hybrid power distribution system.

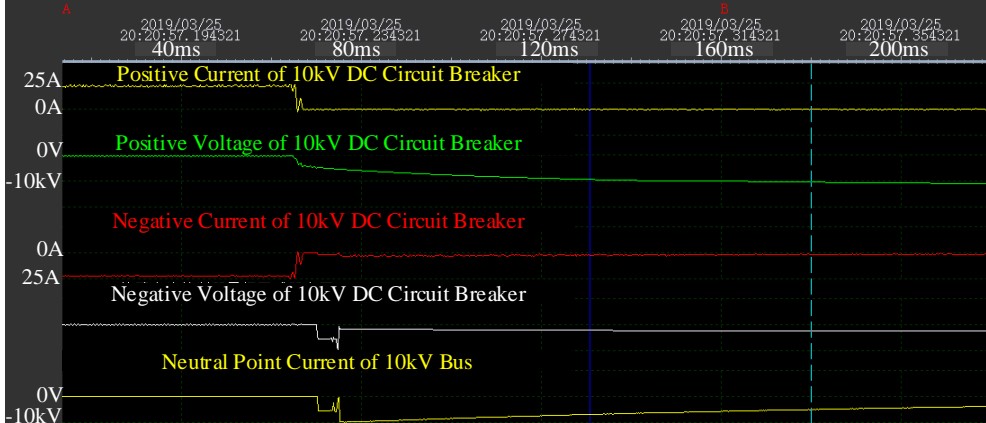

**Figure 12.** Recording of a current breaking test of the AC/DC hybrid power distribution system.

## 6. Conclusions

In this paper, an engineering prototype of the mechanical DC circuit breaker is developed and applied to a AC/DC hybrid distribution network.

(1)　The system structure and system parameters of the AC/DC hybrid distribution network are given, and the technical parameters of the DC breaker are proposed for engineering fault protection requirements.

(2)　The topology and working principles of the mechanical DC circuit breaker are proposed. The engineering prototype of the mechanical DC circuit breaker is developed and the current breaking test is carried out. In particular, the failure mechanism of the breaking current of the mechanical DC circuit is analyzed and verified by tests. The test results show that the mechanical DC circuit breaker can break the 10 kA fault current within 10 ms and withstand 1.5 times overvoltage.

(3)　The engineering prototype of the mechanical DC circuit breaker is installed in the AC/DC hybrid distribution network project, and the breaking test is carried out. The test results show that the mechanical DC circuit breaker has good running condition, and the function and performance meet the engineering design and operation requirements.

Through the prototype development and engineering application of the mechanical DC circuit breaker, the faults of the AC/DC hybrid distribution network can be quickly isolated, thus ensuring safe and reliable operation of the system, and it is of great significance to the optimization of the design and application of mechanical DC circuit breakers.

**Author Contributions:** Conceptualization, L.Q. and Z.Y.; methodology, Y.H.; validation, L.Q., Y.H. and W.Z.; formal analysis, L.Q.; data curation, L.Q.; writing—original draft preparation, L.Q.; writing—review and editing, R.Z.; visualization, L.Q.; supervision, R.Z.; project administration, L.Q.; funding acquisition, X.X.

**Funding:** This work is supported by National Key Research and Development Program (2017YFB0903203), Science and Technology Program of China Southern Power Grid (GDKJXM20172171).

**Conflicts of Interest:** The authors declare no conflict of interest.

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
