# Peer review of "Development and Application of a 10 kV Mechanical DC Circuit Breaker"

_energies, doi:10.3390/en12193615_

Round 1
Reviewer 1 Report
The article presents the development and testing of a dc breaker for 10kV applications with associated experimental results:
The authors should provide some further explanations on the topology of the ac-dc distribution network especially with regards to the ratings of the 10kV dc loads. Section 2 seems to be the article template and not part of the paper. Please remove. The authors should explain the values for the choice of the inductor values. The 2mH, …. 18mH values are rather arbitrary and a per unit explanation for how they are chosen should be given. The authors should provide detailed parameters for their dc breaker including characteristics of the MOV etc The quality of Fig. 3 is very poor and none of the explanations can be seen. It is mandatory for authors to improve the quality of this figure. Table 4 should be further explained. How exactly does it link to the rest of the article? Scales on the y axis on Fig. 11 should be providedAuthor Response
Please see the attachment.

Reviewer 2 Report
Dear Author:
The paper is of the interest to Energies and the designing and experiments results from the prototype is very interesting to see. And the application of the prototype in the hybrid distribution network is appreciated. The novelty of the paper needs to be improved as mechanical DC CB topology has been studied a lot. I wonder if it could be more valuable to focus in details on the application of the hybrid distribution network and the performance analysis.
L72-L84: Section 2 seems not reasonable in the paper, is this part of the template or could you explain it in more details?
L85: Section 3 shows the 'Parameter Configuration of Circuit Breaker'. However, only the inductance of the current limiter is discussed. Could you have more discussion about the other CB parameters like oscillation capacitance and inductance?
L97: the current limiter is within the range of 2mH -18mH, what if the current limiter inductance is larger or smaller?
Table3: please check the current flow direction!
Figure3: please update the figure 3 with vector figure, it is not clear to readers.
L164: 4.2.2. Development of solid-state switch, it shows the disadvantage of the spherical gap but no explanation of the advantage of the solid-state switch. Could you explain more about the advantage of the solid-state switch?
Figure5 is not clear, meanwhile, the left side, the middle and the right side are not clear at all.
L211: 4.3.1 the small current interruption
You explained the high transient voltage across the VI, could you please explain it with equation and show the experimental results in Figure 8? This is a big challenge for the VI.
Figure10: "circuit breaker" not "circuit break"
Figure11 is a screenshot of the oscilloscope, could you zoom in the interruption period and plot the waves in a clear figure?
Round 2
Reviewer 2 Report
Dear Authors,
The test results are not well presented, the readers may be confused with the test results. Please do not show the screenshot of the oscilloscope in the paper.
There have been many research papers on the 10kV DC CBs, please show the novelty of your research work. The paper can be well improved afterward.
Author Response
First of all, I would like to express my heartfelt thanks to the reviewers for their pertinent comments and the editors for their hard work. According to the questions raised by the reviewers, the author of the paper conducted in-depth thinking and analysis to further improve the content of the paper.
This revised description is a detailed response to each question raised by the reviewer. The revised part of the revised manuscript has been changed to red font for review by the reviewer.
Reviewer #2:
1) The test results are not well presented, the readers may be confused with the test results. Please do not show the screenshot of the oscilloscope in the paper.
Reply1:
Thanks to the reviewer for a careful review.
In view of the limitations of the test recording device, the test data cannot be exported, and the waveform cannot be redrawn. Therefore, Figure 11 is a screenshot of the recorded wave.
The new test equipment has been purchased and will be used in subsequent tests.
2) There have been many research papers on the 10kV DC CBs, please show the novelty of your research work. The paper can be well improved afterward.
Reply2:
Thanks to the reviewer's suggestion.
There have been many studies on mechanical DC circuit breakers, and a related review has been introduced in the introduction. The new ideas in this paper on mechanical DC circuit breakers are reflected in two points:
(1) This paper solves the problem of small current breaking of the traditional mechanical DC circuit breaker and carries out test verification. Please see section 3.3.1 for details, and the authors highlight this in the summary.
(2) In this paper, the mechanical DC circuit breaker is applied in the project, and the fault current breaking test is carried out, which provides a reference for the optimal design and operation of the DC circuit breaker. Please see section 4 for details, and the authors highlight this in the summary.